# Experiences of the clinical academic pathway: a qualitative study in Greater Manchester to improve the opportunities of minoritised clinical academics

Chiu-Yi Lin [1], Cinzia Greco,[2,3] Hema Radhakrishnan,[1] Gabrielle M Finn [3], Rachel L Cowen,[3] Natalie J Gardiner[4]

## ABSTRACT

**Objectives** The aim of this study was to explore the barriers and facilitators faced by clinical academics (CAs) in the Greater Manchester region, with particular attention to the experiences of minoritised groups.

**Design** A qualitative study using semistructured interviews and focus groups was conducted. A reflexive thematic analysis was applied to identify key themes.

**Setting** University of Manchester and National Health Service Trusts in the Greater Manchester region.

**Participants** The sample of this study was composed of 43 participants, including CAs, senior stakeholders, clinicians and medical and dental students.

**Results** Six themes were identified. CAs face several barriers and facilitators, some of which—(1) funding insecurity and (2) high workload between the clinic and academia—are common to all the CAs. Other barriers, including (3) discrimination that translates into struggles with self-worth and feeling of not belonging, (4) being or being perceived as foreign and (5) unequal distribution of care duties, particularly affect people from minoritised groups. In contrast, (6) mentorship was commonly identified as one of the most important facilitators.

**Conclusions** Cultural and structural interventions are needed, such as introducing financial support for early career CAs and intercalating healthcare students to promote wider social and cultural change and increase the feelings of belonging and representation across the entire CA pipeline.

## STRENGTHS AND LIMITATIONS OF THIS STUDY

⇒ In-depth interviews with a broad range of participants were conducted to explore the multidimensional barriers faced by clinical academics and the role played by stakeholders in supporting them.

⇒ An innovative qualitative analysis combining thematic analysis and extended case method was used to analyse and interpret the data.

⇒ Only doctors and dentists were interviewed, professionals from other allied health disciplines were excluded.

⇒ The data focus only on the experiences of professionals working in Greater Manchester.

For numbered affiliations see end of article.

**Correspondence to**
Dr Hema Radhakrishnan;
Hema.Radhakrishnan@
manchester.ac.uk

## INTRODUCTION

The diversity profile of clinicians who choose the clinical academic (CA) pathway is not representative of the demographics of either clinical professionals with similar entry qualifications or that of the general population that they serve. Furthermore, the CA data shows that there are systematic inequalities within CA careers as the level of seniority increases.[1] It is evident that the proportion of CAs from minoritised backgrounds is still relatively low in the field. A survey in the UK found that only 33% of CAs are women, ranging from 13% to 58% in different specialities.[2] This also largely reflects women often having lower academic rank and few attaining a leadership role in clinical academia.[3 4] In terms of ethnic profiles, minoritised ethnic CAs are even less represented with only 17% of CAs identifying as Black, Asian or Minority Ethnic backgrounds.[2]

A CA is a qualified healthcare professional who is engaged with clinical responsibilities in a healthcare setting while simultaneously holding an academic position and contributing to either research and teaching, or both. There is a well-defined career pathway for CAs with intercalation, based on clinical experience and academic qualification (https://www.catch.ac.uk/example-medical-clinical-academic-training-pathway). In the British medical context, doctors and dentists share the same pathway.

The diversity in the medical academic workforce is pivotal to increase innovation in healthcare, deliver high-quality patient care and provide visible role models and mentoring support for students and early career professionals, especially for those from minoritised ethnic backgrounds.[5 6] This also brings benefits to public health such

as reducing health inequalities and building long-term trustful relationship with disadvantaged communities.[7 8]

Various challenges have already been recognised for CAs, such as limited funding, financial insecurity, limited CA positions and the demanding workload of a clinical role.[9–13] For CAs from minoritised backgrounds, those challenges could be compounded. For example, for female CAs, social expectations around family commitment and parenting responsibilities could result in their being unable to pursue CA careers.[9 14] In some cases, parental leave is limited by gender-based discrimination and sexism in the work culture.[9] Barriers such as the under-representation of ethnic minority colleagues in workplaces, a lack of cultural competency and trust from people at senior levels contribute to the challenges for CAs from ethnic minority groups.[13 15]

Previous work has highlighted the value of intercalation for healthcare students to the diversity of the CA pipeline.[11 16] Interrupting professional training to develop research skills at the early career stage of future healthcare professionals is critical to widen access and participation for under-represented groups in clinical academia. The removal of educational achievement scores from the allocation process for foundation posts is a cause for alarm, as it will have far-reaching consequences for student uptake of intercalation and amplify the inequities in the CA pipeline.[17] Significant investment is required to prevent further inequity at early career stages, such as funding and scholarships to students from disadvantaged backgrounds.

Increasing the diversity of the CA workforce is emphasised in policies internationally.[18] In the UK, the National Health Service (NHS) Race Equality Action Plan presented a mandatory workforce race equality standard.[19 20] According to Kline *et al*,[21] in Northern England (which includes Greater Manchester), Ethnic minority representation at board and very senior management level was significantly lower than in the overall NHS workforce and also in the local communities served.[21] Therefore, some regional actions have been taken to tackle the situation. The Manchester Foundation Trust introduced 'the Trust's Equality, Diversity and Inclusion Strategy 2019–2023' to ensure a representative and supported workforce.[22] There has been an improvement in diversity of CAs (gender and ethnicity) both in UK academic institutions and in the NHS.[1 23] While there has been an improvement in the representation of CAs, more work still needs to be done to attract and importantly sustain a diverse population of CAs.

It is critical to understand the negative and positive experience of doctors and dentists from minoritised backgrounds, the barriers they experience and areas of support and institutional good practice in removing any systemic barriers. This data will inform recommendations at a local level, which will also be applicable to a wider UK context.

## METHODS
### Design
A qualitative study using semistructured interviews and focus groups was conducted to explore the experiences of the CA doctors and dentists in Greater Manchester, with a particular focus on those identifying as being from minoritised backgrounds (including but not limited to gender, race/ethnicity and/or sexual orientation).

### Setting and samples
Theoretical and snowball sampling were employed to recruit CA (doctors and dentists, in research-focused, teaching-focused and research and teaching posts) and medical and dental students (years 3–5) from the University of Manchester. Clinicians in Greater Manchester, key senior stakeholders and gatekeepers to CA and clinical posts in Greater Manchester were also recruited. The aim of this study was to understand the experiences of people at different stages of the CA career pathway. Therefore, a wide range of participants from students to senior academics/clinicians were included. This generated diversity of thought in developing an understanding of the barriers and enablers that influenced the decision to pursue a CA career. The aim of theoretical sampling is to 'generate theory whereby the analyst jointly collects, codes and analyses his data and decides what data to collect next and where to find them in order to develop his theory as it emerges'.[24] Snowball sampling is a technique where existing participants provide referrals of potential participants interested in taking part in the study.[25] In order to capture the diversity of experiences, participants were recruited from a wide range of health settings, at various stages of their careers and with an emphasis to include minoritised profiles.

### Data collection
Potential participants were identified through the members of established medical education networks, CA training and academic networks of the research team and the snowball sampling approach involving already interviewed participants. Posters and social media platforms (such as Twitter and Faculty newsletters at the University of Manchester) were used to advertise the study. Any potential participants who were interested in taking part were able to contact the authors directly. All potential participants were provided with an online information sheet and gave fully informed consent before and during participation in the research. The research was designed to take into account the busy schedule of CAs, clinicians and stakeholders, and the participants were given the option to choose between individual semistructured interviews and focus groups. All participants, except the students, chose the individual interview because it was easier to fit in their schedule. Interview topic guides for semistructured individual interviews and focus groups are available as online supplemental file 1.

Previous studies have shown information power[26] to be important when the intention is to collect a rich account

of experiences of participants. This means that the more information the sample holds (such as that provided with the in-depth interviews in the present study) the desired information power is achieved with fewer participants. The concept of information power focuses on how to accurately represent participants views, generate in-depth data and cover the variability of relevant events.[26] This study was guided by the principle of information power, as the research team aimed to include participants from different stages of the CA career pathway.

Both focus group and individual interviews were carried out virtually through Zoom (V.5.13, Zoom Communications) or Microsoft Teams (V.1.5, Microsoft), lasting between 17 and 61 min. The interview topics included experiences of the CA career pathway or stakeholder perspective, barriers/facilitators encountered, especially for people from minoritised groups and experiences of local research cultures. Brief field notes were taken for all interviews and focus groups to facilitate data analysis and identify the initial coding. All interviews and focus groups were conducted by first two authors (C-YL and CG). Data collection was undertaken from September 2022 to February 2023.

### Data analysis

All interviews and focus groups were digitally recorded, transcribed and analysed following Braun and Clarke's reflexive approach to thematic analysis.[27] Further, to consider existing qualitative research on the careers of CA and the barriers faced by minoritised groups, the interpretation of the results was integrated with the principle of 'reconstruction' from the extended case method proposed by Burawoy, in which anomalies in relation to an established theoretical corpus are considered and integrated in the analysis through the addition of new themes.[28]

### Patient and public involvement

This research did not involve patients but only medical personnel and students; patients were not involved in the design, or conduct, or reporting, or dissemination plans of this research.

### RESULTS

The study aimed to explore the experiences, barriers and facilitators that CAs (doctors and dentists) encounter in pursuing a CA career. A total of 43 participants took part in either an interview (37 CA, clinician and stakeholder participants) or a focus group for this study (6 medical or dental student participants). In this study, 21 men and 22 women with a wide range of roles participated (19 CAs, 5 clinicians, 13 stakeholders and 6 medical or dental students). The 6 students were all in their 20s, the other participants were aged between 30 and 70 years (9 were 30–39; 7 were 40–49; 18 were 50–59 and 3 were 60+). Of the 43 participants, 32 were White, 9 were Asian and 2 were Black. The semistructured interview framework

**Table 1** Themes and subthemes

| Theme | Sub-theme |
|---|---|
| A highly competitive pathway that gives limited job security | Financial challenges |
| | Jobs insecurity |
| | Limited funding |
| Balancing two high-workload careers | Heavy workload |
| | Limited time for conducting research |
| Lack of representation and role models as limits to belonging | Diverse role models |
| | Support from peers with a similar background |
| Being foreign or perceived as such | A feeling of belonging |
| | A language barrier |
| Childcare and gender inequalities | Maternity leave |
| | Parenting responsibility |
| Support and recognition through mentorship | Supportive mentorship from peers with similar experience |
| | Useful advice from mentors |

focused on both the barriers and facilitators for minoritised groups and also general barriers and facilitators applicable to all CA.

Six main themes were generated—theme 1: a highly competitive pathway that gives limited job security; theme 2: balancing two high-workload careers; theme 3: lack of representation and role models as limits to belonging; theme 4: being foreign or perceived as such; theme 5: childcare and gender inequalities; theme 6: support and recognition through mentorship. Of these themes, the first two themes describe barriers common to all CAs, themes 3–5 describe barriers that impact in particular participants from minoritised groups, while theme 6 describes one of the main facilitating factors for all CA. The details of themes and subthemes are summarised in table 1.

### Theme 1: a highly competitive pathway that gives limited job security

Participants expressed a strong feeling of insecurity in relation to their own career as a CA, describing the CA pathway as more precarious when compared with a clinical career. Participants, regardless of their level of seniority, recognised that there is fear while holding fixed-term positions of losing one's job and of being unable to control how long CA employment would be based on the success rate of obtaining grants and the duration of the grants.

Several participants mentioned the *financial challenges* of the CA pathway, including precarity and uncertainty linked to fixed-term contracts and *limited funding*. It was seen as problematic that CAs needed to secure funding to continue in their career paths, particularly in the early stages of their career. One gatekeeper said:

There is a lot of career uncertainty in academic life. I think it's becoming increasingly so since, say, in the last decade or so, because of funding difficulties in higher education. Determined junior members of staff may start out enthusiastically trying to cut a path in academic life, but I think they become frustrated at the lack of security in their job and it's all too easy to feel that you could move into a less rigorous discipline. [G04]

Feelings that they were at risk of *losing their jobs* after the completion of the academic clinical lecturer (ACL) training were widely reported by the participants. One said:

I suppose the key step is if you're at the ACL level either with a formal ACL or trying to get your first independent fellowship, if that doesn't get funded by the time your ACL ends then just to earn a living you have to go back to the NHS, and then I think it's quite difficult. You've got to be very driven to be able to drive forward a fellowship application whilst you're still working full time in the clinical arena. [G07]

One of the privileged routes to enter the CA pathway is through fellowships, and *funding opportunities* for fellowships are extremely competitive. One participant mentioned:

There's a sort of a wish that everybody will go and get a very prestigious MRC [Medical Research Council] or Wellcome Trust fellowship, but actually there's not enough of those to go around. [G06]

The participants who obtained the most prestigious fellowships and had straightforward careers (eg, transiting from specialisation to academic clinical fellow and then lecturer progressing on to a permanent CA position) also reported being concerned or anxious about finding their next job or securing their next grant in academia. The participants felt that continuous and secure employment was increasingly difficult to find. Therefore, uncertainty of subsequent employment and funding was also described as a major barrier to staying in the field. For example:

As opposed to research, where you literally, if you don't have the next thing [research job], it's like where is your next pay cheque coming from. So, also making that not a thing for people who are specifically research. I don't know how you would do that. It's just, yeah. I know there's no easy answer to some of these things. [CA07]

While all participants were keen to pursue CA career pathways, the *uncertainty of employment* as a CA emerged as a constant source of anxiety. As a result, the participants felt that they needed to prioritise and maintain an income to meet their basic living costs. A number of participants believed that a purely clinical career was more attractive

due to its stability and clearly defined clinical progression route. One participant also explained:

Academia does not support a family life, whereas the NHS does so. And when I speak to my peers who are in academia, that's what they say: they are just not supported. And the way that academia works, is that you're going from contract to contract and grants, no stability at all. So, I'm glad I'm not an academic for that reason. [G08]

The importance of securing and *sustaining funding* was noted as a key challenge by a number of the participants. Supporting early and mid-career CA fellows through internal and external bridging funds while they are preparing for their next grant application was identified as a facilitator for career progression. One mid-career CA shared the following experience:

These things [funding] are very competitive. I was very fortunate: I got the \*\*\* Clinical Training Fellowship, the first I applied for, and I was successful, so that was fortunate. The Intermediate Fellowship took me a few goes and at the time there was no… there were no schemes to support clinical academics in that intermediate phase. And I think it's a bit better now because there's the NIHR clinical lectureship scheme that provides some reaching [bridging] support, because you can't jump straight from a clinical train to a PhD and straight to an intermediate fellowship. So, I was very fortunate to be working in an environment where there was funding to support me, um, but that was a challenge. [CA18]

### Theme 2: balancing two high-workload careers

CA careers can offer the benefits and rewards of making a difference to patients and having the space to pursue research projects. However, the challenge of how to *balance an academic role with a clinical career* was one of the common barriers mentioned:

And I say to all clinicians who express an interest in academic work that everything is possible, but that you are embarking upon a career that has a clinical component and academic component, and you will be juggling, and it will affect the intensity of your work. So, people have the awareness that this isn't gonna be the easiest route. [G11]

The workload for both the clinical and academic roles often goes beyond what is anticipated based on the CA job plan. Effective line management and responsive workload management between both roles can reduce stress and improve satisfaction while meeting the demands of both roles. From participants' reports, *workloads* go beyond the official hours both in the clinic and in academia. In addition, balancing research, teaching and clinical roles was a common challenge:

Two-and-a-half days were taken up by my clinical work, which is the same as was, and two days were taken up

for teaching, so it's not quite accurate to say I had no research time, but I had half a day of research time in that five days and when you are a clinical academic you need to do lots of things to build up your name, especially when you're starting. You need time to go for grants. You need time to write papers. You need time to network and do increasing amounts and gain momentum. [CA09]

A common difficulty was the feeling of being considered as not doing enough work in the clinic compared with full-time colleagues, with the research work in some cases being considered less strenuous than the clinical work that CAs are expected to do in the same measure as full-time colleagues. For junior clinicians, this could mean being potentially seen as someone not acquiring the necessary clinical skills, as in the case of this junior CA:

If we are not in theatre learning, we won't acquire our skills. Particularly from a clinical perspective it requires a lot of flexibility from trainers. These are surgeons, they just live, breathe, eat surgery. I am there 50% of the time. Sometimes they see me, sometimes they don't, and then they ask me 'are you here tomorrow?' You say 'no, I am not here'. You feel inadequate, you feel like you are not giving 100%. [CA06]

### Theme 3: lack of representation and role models as limits to belonging

Barriers linked to systemic forms of discrimination can act on a more personal level, undermining a person's sense of self-worth and well-being. Sexism, racism and other forms of discrimination that individuals experience throughout their life can impact their work life.

But as an ethnic minority you feel those, you have grown up with them and some of them are not overt. Prejudices are, for me, I can usually sense them rather than someone […] overtly saying no […] You have heard it so many times, you kind of just think, right, okay, I am going to go and knock on a different door. I would say those things are the things that made me constantly more think I have to prove myself, I have to prove myself. [CA06]

The importance of representation and a feeling of belonging is crucial for a diverse and dynamic workforce. However, it is common for some specialties, research fields and/or senior leadership positions to be occupied by White British men. Many of the women participants have, for example, described leadership positions within academia as closed circles. Indeed, many of the women who did not think that their gender has negatively impacted on their career recognise that some higher roles are much more male dominated.

We still live in a world where they [men] dominate senior academic and senior clinical and senior college and senior government type positions, and so they make a lot of decisions. I think that they occupy that space more than women, and that is levelling up all the time. I don't think they've ever stood in my way. [CA11]

The lack of *diverse role models* and peers with whom it was possible to identify was discussed, along with the feeling of being out of place in different settings.

If I go to a meeting of researchers […] I might be the only South Asian in the room. That's when I feel like a minority; I never feel like a minority in clinical meetings. [CA09]

And even less so when you're Black and woman, there's even less of that [ethnic minority women in CA] And sometimes it's like, oh…can you do this? [CA07]

Greater Manchester is arguably one of the most diverse areas of the UK. The importance of diversifying the CA pipeline was recognised in the Trust's Equality, Diversity, and Inclusion Strategy 2019–2023 (introduced by Manchester University NHS Foundation Trust),[22] which led to actions such as improving diversity of interview panels. Some interviewees reported positive experiences, such as this female ethnic minority CA:

So, I went and did the interview and my interview panel actually was a group of all women, all different ethnicities and, I don't know, immediately I just felt way more at ease with them. I felt like I could be myself. We spent 45 minutes talking about me and my career goals and what I wanted and they really…I felt like they really got to know me and they gave me that time and respect to open up and answer questions. [CA04]

### Theme 4: being foreign or perceived as such

Along with the difficulties linked to race and ethnicity, some of the barriers discussed by the interviewees were more specifically linked to not being British born or not being British citizens. Having a foreign accent was recognised as a marker of not being British born, and potentially non-British trained. As a result, CAs with the same ethnicity could have very different experiences. As one participant discusses:

The other thing I think is something to do with how well you can speak and whether you have an unusual accent, which some people do, sometimes that accent can be taken as a bias against subconsciously, I feel. It hasn't happened to me, but I have seen other people who are very capable, but with a strong accent […] maybe could have done better. [C04]

*Language barriers*, even if very modest, were also recognised as potentially creating a disadvantage in publishing and therefore building esteem and an academic career. Others underlined the role that discrimination can play, such as the peer review of grants and articles. Interviewees

have underlined how in these cases the doubt remains whether a non-English name might negatively influence the outcome.

## Theme 5: childcare and gender inequalities

The most discussed barriers experienced along gender lines were linked to maternity and childcare and return to work following a period of maternity leave was described as difficult by a number of interviewees, particularly where the expectation was that work can resume without any disruptions.

> The one thing [a supervisor] said to me was 'you just need to make sure that your childcare is bomb-proof'. So, I don't know how you can manage to have childcare that is completely perfect. [G11]

Although maternity leave is recognised when accounting for career breaks in the academic CV, a number of interviewees felt that interruptions for maternity were not taken into account when considering the track record of candidates for promotion and career progression. *Maternity* was recognised as a source of discrimination even by women who did not have children. One stakeholder observed:

> I think [that] for women [it is difficult] to progress because of the time barrier. If you're then having children and you're trying to juggle a clinical academic career, it's almost like you have two full time jobs… So, we lose a lot of women, and that's why you tend to see a lot of male clinicians who are then on the research path. [G12]

*Childcare* was further linked by several participants to barriers for CA careers. A number of interviewees discussed how childcare continues to largely fall on women, and how it is difficult to reconcile the already high workload of clinic, research and/or teaching with parenting. The difficulty of managing the different workloads combined with the demands in terms of flexibility and potential need to relocate linked to the academic career. As a result, a number of interviewees observed how a purely clinical career was more attractive as it was more stable with less workload outside hospital hours, and how this might be an incentive to abandon the CA pathway.

## Theme 6: support and recognition through mentorship

Mentorship was identified as a main facilitator for CA career development. Mentorship relationships were mostly built through personal contacts, professional networks or formal mentorship programmes. Typically, the participants were more likely to seek mentorship from people who have the same or similar life experiences and were successful CAs. The participants believed that supportive mentorship could facilitate exploring their research interests, building their portfolio and expanding their professional networks. Mentors were able to provide deep insight into CA career pathways, for example:

> The person who's helped me the most was my PhD supervisor. He is a clinical academic; he interviewed me the first time round and didn't give me the fellowship, but encouraged me to seek other opportunities. He knew… because he'd been through it himself, he was aware of the trajectory, he was aware of what was needed, he really helped me in terms of how to write a fellowship application, how to sell yourself, really, which I think is something that I struggle with. So he's been very instrumental in helping my career, definitely, and seeing that I potentially have a future in this. [CA03]

Participants who self-identified as earlier career researchers pointed out the importance of *senior and peer mentors* in CA career pathways. Beyond the academic support, some participants also noted that mentors are pivotal when they encounter challenges in the field. The benefits of mental or emotional support from these mentors were reported. Based on their advanced knowledge or experience, mentors were in a good position to offer *advice on CA career pathways*. The participants felt they were protected: in a safe place. They believed these mentors could help them to develop into professional and independent CAs, for example:

> I had a fantastic mentor […] and she kind of took me under her wing. She could see I felt a bit lost. I didn't get the training number. I was, you know, five hours away from my family and she kind of just gave me this motivation. And some focus to say, 'Right. You're next. I know it's difficult for you up here and you're lonely, but you need to focus on the next step.' And she was pivotal really, in that she really believed in me. Like, really, for somebody, I didn't know very well. I don't know. She had this incredible ability to make me quite confident in my skills. [CA06]

## DISCUSSION

The participants to our research project form a diverse group including CAs, clinicians, students and senior stakeholders, which has made it possible to understand a wide range of difficulties and problems impacting decision-making and career choices. Six themes have been identified, with two describing general barriers experienced by CA, three capturing further the complex experiences of minoritised groups, and one identifying an important facilitator.

This project has corroborated previous research[9 10] in showing the difficulties in managing two demanding careers and in finding a balance for activities with a significant workload that are difficult to scale to part-time positions, and how this creates challenges for career advancement, especially for people with childcare responsibilities.

Both this study and previous research reported that various challenges such as job insecurity and heavy

workload could be key barriers to pursuing a CA career.[9–13] Several new elements emerge from the analysis of our data. Among the general barriers, our participants mentioned recent changes in the career pathway, including the reduction of available positions, coupled with the expectation of obtaining prestigious fellowships before advancing in CA careers.

The participants highlighted the degree to which obtaining such fellowships requires time, support from experienced colleagues and availability of bridge funding to allow time to reapply in case of rejection. While available in the context studied, such resources are unequally distributed and require in turn a significant time and financial investment for an uncertain outcome. In particular, support from peers and colleagues, particularly at a senior level, was recognised as a key facilitator to tackle these challenges, both in our study and in the previous literature.[10 11] A recent systematic review of the existing interventions to support CA careers for doctors and dentists found that protected research time could enhance CA retention and career progression.[11] In addition, to bolster the feeling of belonging for minoritised CAs, representation and inclusion is crucial.

One of the main individual barriers is the ways in which being minoritised hinders one's confidence. Ethnic minority participants reported episodes of racism, in their careers, but further discussed how experiences of discrimination and racism suffered in other spheres of life can have an impact also on their self-worth in a professional setting. A strong sense of being an 'outlier' is still commonly reported from women and minority populations in the research environment in the literature.[9 13–15] This might explain the feeling of unbelonging that further erodes confidence. For example, women participants perceived men occupying the majority of the senior and management positions as disempowering. Despite improvement in female representation in leadership positions, gender imbalance can impact career progression through the phenomena already identified as glass ceiling and/or sticky floors.[9 10] Ethnic minority participants have reported similar experiences, with further issues such as having a name perceived as foreign and being potentially disadvantaged in being evaluated for publications and grants.

Taken together, these data highlight the range of systemic barriers experienced by minoritised CA. These ranged from how having an accent was perceived as limiting progression to lack of role models, difficulties in balancing childcare commitments and two highly demanding roles. The study also highlighted key enablers through bridging financial support, grant writing support and recognising lived experience and supporting career progression through mentorship. These findings will help inform individuals, senior leaders and organisations of the facilitators and barriers that impact on CA career progression and help policy development for those leading on CA training programmes and delivery of a targeted equality, diversity and inclusion strategy.

There are limitations to this study: the fact that only a quarter of the respondents were from minoritised groups should be taken into account in the analysis of the results. Further studies with wider and more diverse samples might be helpful to enhance transferability in qualitative research. Meanwhile, as one aim of this study was to understand the experiences of minoritised groups, the composition of the research team should also be considered. Our team is diverse in terms of ethnicity and national origin, but is entirely female, which may have had a role in the collection and analysis of the data.

## CONCLUSIONS

The study showed that there are several barriers that current and prospective CAs face. It has also shown that barriers and difficulties are increased for participants from minoritised groups. To remove these barriers, cultural and structural interventions are needed. On the one hand, it is important to identify forms of financial support for early career CAs, in order to make the CA pathway more financially stable. On the other hand, it is important to encourage wider social and cultural changes that can reduce the feelings of unbelonging that women and ethnic minority participants in particular have expressed. These findings and recommendations are of relevance for senior leaders and organisations involved in CA training and more general equality, diversity and inclusion and careers strategies.

**Author affiliations**
[1]Division of Pharmacy and Optometry, School of Health Sciences, Faculty of Biology, Medicine and Health, The University of Manchester, Manchester, UK
[2]Centre for the History of Science, Technology and Medicine, The University of Manchester, Manchester, UK
[3]Division of Medical Education, School of Medical Sciences, Faculty of Biology, Medicine and Health, The University of Manchester, Manchester, UK
[4]Division of Diabetes, Endocrinology & Gastroenterology, School of Medical Sciences, Faculty of Biology, Medicine and Health, The University of Manchester, Manchester, UK

**Acknowledgements** Thanks to Professor Colette Fagan and Professor Adam Danquah, University of Manchester for helpful discussions during the course of this project. Thanks to Megan Brown, University of Newcastle for her help with ethics. Thanks to clinicians, clinical academics, students and stakeholders who participated or facilitated this study.

**Contributors** HR and NJG conceived of and led the study as principal investigators and act as guarantors for the study. C-YL and CG conducted data collection and data analysis. GMF, HR and NJG contributed qualitative data analysis. All authors (C-YL, CG, HR, GMF, RLC and NJG) made contributions to the manuscript and reviewed the final submission.

**Funding** This work was supported by University of Manchester and Wellcome matched funding reference 204796/Z/16/Z ISSF—Wellcome ISSF 3—EDI.

**Competing interests** None declared.

**Patient and public involvement** Patients and/or the public were not involved in the design, or conduct, or reporting, or dissemination plans of this research.

**Patient consent for publication** Not applicable.

**Ethics approval** This study involves human participants and was approved by The University of Manchester ethics committee (Reference: 2022-13791-23819). Participants gave informed consent to participate in the study before taking part.

**Provenance and peer review** Not commissioned; externally peer reviewed.

**Data availability statement** No data are available. The interview and focus group data used in this study are not publicly available due to integrity statements in the ethics approvals.

**ORCID iDs**
Chiu-Yi Lin http://orcid.org/0000-0002-0302-6535
Gabrielle M Finn http://orcid.org/0000-0002-0419-694X

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
