## [Reviewer comments · BMJ Open]

ARTICLE DETAILS

TITLE (PROVISIONAL)	Experiences of the clinical academic pathway: A qualitative study in Greater Manchester to improve the opportunities of minoritised clinical academics.
AUTHORS	Lin, Chiu-Yi; Greco, Cinzia; Radhakrishnan, Hema; Finn, Gabrielle; Cowen, Rachel; Gardiner, Natalie

VERSION 1 – REVIEW

REVIEWER	Nimmons, D. UCL
REVIEW RETURNED	20-Sep-2023

GENERAL COMMENTS	Thank you for asking me to review this paper. Overall, I think the topic is interesting and valuable. However, I have a few comments and suggestions. Introduction It would be good to have a description and diagram showing an example of an academic career path in the UK, to help provide context. For example, intercalation to academic foundation to ACF to PhD to ACL. Methods Sample size in qualitative research is more guided by the concept of 'information power' instead of saturation. Information power encourages researchers to consider the richness of the dataset as opposed to the sample's size and is widely recommended to inform sample size, in preference to the concept of 'data saturation' which is difficult to achieve. Something around this should be added to the methods. In the strengths and limitations the authors mention 'Extended case method' but I cant see this mentioned in the methods. Results It would be useful to have a table showing participant characteristics. Theme 5 is interesting. The authors say maternity was a source of discrimination, do they have a quote that supports this? Discussion There is limited comparison to existing literature. Please expand on how the results relate to previous studies. It would be helpful to have a section on recommendations.
---

	Limitations Statistics aren't used in qualitative research when it comes to sample size. I would remove the sentence that mentions statistics. 43 is a large sample but I agree there are not many ethnic minorities, especially participants who identify as Black. This could be explored further in future research.
--	---

REVIEWER	Midik , Ozlem Ondokuz Mayis Universitesi
REVIEW RETURNED	06-Oct-2023

GENERAL COMMENTS	Thank you for this study . I think the design and sampling of this qualitative study needs to be revised. The sample is too large and does not focus on one context. There are doctors and dentists at the same time and students at the same time. The reason for this is not fully understood. The experiences of these people with different characteristics and working conditions will be different from each other. If it could reveal this difference, maybe it would be understandable... but it was not. Individual and focus group interviews were conducted at the same time. It is not clear who was interviewed individually and who was interviewed in a focus group. Why were two different tools used? What are the research questions? It could not be traced from the article. It belongs to the sampling mentioned in lines 19-33 and is included in the findings section. What does 'theoretical and snowball sampling' mean? Themes and codes are not clearly traceable in the article. It is recommended to address the purpose, design and methodology and rewrite the article.
---

VERSION 1 – AUTHOR RESPONSE

Reviewer: 1

Dr. D. Nimmons, UCL

Comments to the Author:

Thank you for asking me to review this paper. Overall, I think the topic is interesting and valuable.

However, I have a few comments and suggestions.

Accepting comments:

Introduction

It would be good to have a description and diagram showing an example of an academic career path in the UK, to help provide context. For example, intercalation to academic foundation to ACF to PhD to ACL.

[Reply] Thank you for your comment. We have added a paragraph and a link to an infographic to illustrate the clinical academic career path in the UK (page 3). This reads as follows:

“A clinical academic is a qualified healthcare professional who is engaged with clinical responsibilities in a healthcare setting whilst simultaneously holding an academic position and contributing to either research and teaching, or both. There is a well defined career pathway for clinical academics with intercalation, based on clinical experience and academic qualification (<https://www.catch.ac.uk/example-medical-clinical-academic-training-pathway>). In the British medical context, doctors and dentists share the same pathway.”

Methods

Sample size in qualitative research is more guided by the concept of 'information power' instead of saturation. Information power encourages researchers to consider the richness of the dataset as opposed to the sample's size and is widely recommended to inform sample size, in preference to the concept of 'data saturation' which is difficult to achieve. Something around this should be added to the methods.

[Reply] We have added a paragraph in the methods section (page 6). This reads as follows:

“Previous studies have shown Information power²⁶ to be important when the intention is to collect a rich account of experiences of participants. This means that the more information the sample holds (such as that provided with the in-depth interviews in the present study) the desired information power is achieved with fewer participants.

The concept of information power focuses on how to accurately represent participants views, generate in-depth data, and cover the variability of relevant events.²⁶”

In the strengths and limitations the authors mention 'Extended case method' but I cant see this mentioned in the methods.

[Reply] We have revised the method section and added more information about this (page 6). This reads as follows:

“All interviews and focus groups were digitally recorded, transcribed and analysed following Braun and Clarke's reflexive approach to thematic analysis²⁷. Further, to consider existing qualitative research on the careers of CA and the barriers faced by minoritised groups, the interpretation of the results was integrated with the principle of 'reconstruction' from the extended case method proposed by Burawoy, in which anomalies in relation to an established theoretical corpus are considered and integrated in the analysis through the addition of new themes²⁸.”

Results

Theme 5 is interesting. The authors say maternity was a source of discrimination, do they have a quote that supports this?

[Reply] We have added a quote to this section (Please see the results section, page 14). This reads as follows:

“One stakeholder observed:

“I think [that] for women [it is difficult] to progress because of the time barrier. If you're then having children and you're trying to juggle a clinical academic career, it's almost like you have two full time jobs... So, we lose a lot of women, and that's why you tend to see a lot of male clinicians who are then on the research path.” [G12]”

Discussion

There is limited comparison to existing literature. Please expand on how the results relate to previous studies.

[Reply] Thank you for your comment. We have revised our Discussion and strengthened the links to existing literature. (Please see the discussion section, pages 16-17).

It would be helpful to have a section on recommendations.

[Reply] Given the limits in terms word count, we have included the recommendations in the conclusions section (page 18). This reads as follows:

“The study showed that there are several barriers that current and prospective CA face. It has also shown that barriers and difficulties are increased for participants from minoritised groups. To remove these barriers, cultural and structural interventions are needed. On the one hand, it is important to identify forms of financial support for early career CA, in order to make the CA pathway more financially stable. On the other hand, it is important to encourage wider social and cultural changes that can reduce the feelings of unbelonging that women and ethnic minority participants in particular have expressed. These findings and recommendations are of relevance for senior leaders and organisations involved in clinical academic training and more general in equality & diversity and careers strategy.”

Limitations

Statistics aren't used in qualitative research when it comes to sample size. I would remove the sentence that mentions statistics. 43 is a large sample but I agree there are not many ethnic minorities, especially participants who identify as Black. This could be explored further in future research.

[Reply] We have removed the sentence from the section.

Rejecting comments:

Results

It would be useful to have a table showing participant characteristics.

[Reply] Considering that the participants come from a very limited population in a well-defined geographic area, providing a further breakdown of participant characteristics would risk making the interviewees identifiable. We have included some participant characteristics including age, gender, and ethnicity at the beginning of the results section.

Reviewer: 2

Dr. Ozlem Midik , Ondokuz Mayis Universitesi

Comments to the Author:

Thank you for this study .

Accepting comments:

I think the design and sampling of this qualitative study needs to be revised. The sample is too large and does not focus on one context. There are doctors and dentists at the same time and students at the same time. The reason for this is not fully understood. The experiences of these people with different characteristics and working conditions will be different from each other. If it could reveal this difference, maybe it would be understandable... but it was not.

[Reply] Thank you for your comments. A paragraph and a link to an infographic have been added to give a more detailed idea about academic careers in the UK (please see response above to reviewer 1). In the UK, doctors and dentists share the same academic career path. Therefore, we recruited both doctors and dentists. One recent systematic review (Raine et al., referenced in the paper) also examined interventions to support clinical academic careers for doctors and dentists.

Individual and focus group interviews were conducted at the same time. It is not clear who was interviewed individually and who was interviewed in a focus group. Why were two different tools used? What are the research questions? It could not be traced from the article.

It belongs to the sampling mentioned in lines 19-33 and is included in the findings section.

[Reply] Individual semi-structured interviews and focus groups were offered to all participants so that they were free to choose a format that suits the individual best. As expected the focus group option was only chosen by the student participants whilst all others chose individual interviews. This has now been elaborated in the methods and results sections as follows:

“The research was designed to take into account the busy schedule of clinical academics, clinicians, and stakeholders, and the participants were given the option to choose between individual semi-structured interviews and focus groups. All participants, except the students, choose the individual interview because it was easier to fit in their schedule.”

“A total of 43 participants took part in either an interview (37 clinical academic, clinician, and stakeholder participants) or a focus group for this study (6 medical or dental student participants).”

What does 'theoretical and snowball sampling' mean?

[Reply] We have added more information about the sampling techniques (page 6). This reads as follows:

“The aim of theoretical sampling is to “generate theory whereby the analyst jointly collects, codes and analyses his data and decides what data to collect next and where to find them in order to develop his theory as it emerges”²⁴. Snowball sampling is a technique where existing participants provide referrals of potential participants interested in taking part in the study²⁵.”

Themes and codes are not clearly traceable in the article.

[Reply] We have added a Table 1 summarising themes and subthemes (*italicise the text*) to clarify this aspect (page 7).